# Putative mRNA Biomarkers for the Eradication of Infection in an Equine Experimental Model of Septic Arthritis

**DOI:** 10.3390/vetsci11070299

**Published:** 2024-07-02

**Authors:** Roman V. Koziy, José L. Bracamonte, George S. Katselis, Daniel Udenze, Shahina Hayat, S. Austin Hammond, Elemir Simko

**Affiliations:** 1Department of Veterinary Pathology, Western College of Veterinary Medicine, University of Saskatchewan, Saskatoon, SK S7N 5B4, Canada; 2Department of Large Animal Clinical Sciences, Western College of Veterinary Medicine, University of Saskatchewan, Saskatoon, SK S7N 5B4, Canada; jlb923@usask.ca; 3Canadian Centre for Rural and Agricultural Health, Department of Medicine, College of Medicine, University of Saskatchewan, Saskatoon, SK S7N 2Z4, Canada; george.katselis@usask.ca; 4Next-Generation Sequencing Facility, Cancer Research Cluster, University of Saskatchewan, Saskatoon, SK S7N 5E5, Canada

**Keywords:** horse, septic arthritis, mRNA, biomarker, eradication of infection

## Abstract

**Simple Summary:**

Joint infection, or septic arthritis, is an important disease in horses and requires aggressive treatment to eradicate the infection. Using current diagnostic methods, it is difficult to determine when the infection has been eliminated following treatment, leading to often prolonged antimicrobial therapy, which may be unnecessary and is associated with side effects and antimicrobial resistance. Thus, better markers of eradication of joint infection are needed. In this study, a class of molecules called messenger ribonucleic acid (mRNA) in synovial fluid was investigated for potential biomarkers of the eradication of infection in an experimental model of equine septic arthritis using transcriptomics methods. Transcriptomics data were also compared to our previously published data on putative protein biomarkers of the eradication of joint infection and to an mRNA biomarker panel used to differentiate septic from non-septic shock in humans. Eight mRNAs were identified that were at least three times increased in horses with active septic arthritis compared to horses post-eradication of infection after treatment and horses with non-septic synovitis. The presence of mRNAs corresponding to our previously reported protein markers of eradication of joint infection and to the validated mRNA biomarker panel detecting sepsis in humans was also confirmed. Further investigation of mRNAs as a source of potential markers of the eradication of joint infection in horses is needed.

**Abstract:**

Septic arthritis (SA) in horses has long-term health implications. The success of its resolution hinges on the implementation of early, aggressive treatment, which is often sustained over a prolonged period. Common diagnostic methods do not allow for the reliable detection of the eradication of joint infection. A potential alternative is the discovery and characterization of mRNA biomarkers. The purpose of this study was to identify potential mRNA biomarkers for the eradication of joint infection in equine SA and to compare their expression with our previously published proteomics data. In addition, the transcriptomics data were compared to the mRNA biomarker panel, SeptiCyte Lab, used to distinguish sepsis from non-septic shock in humans. A comparative transcriptomics analysis of synovial fluid from the SA joints of five horses with active infection and subsequent post-treatment eradicated infection in the same joints and five horses with non-septic synovitis was performed. Eight novel mRNA transcripts were identified that were significantly upregulated (>3-fold) in horses with active SA compared to horses post-eradication of infection after treatment and horses with non-septic synovitis. Two proteins in our proteomics data corresponded to these mRNA transcripts, but were not statistically different. The transcripts used in the SeptiCyte test were not differentially expressed in our study. Our results suggest that mRNA may be a useful source of biomarkers for the eradication of joint infection in horses and warrants further investigation.

## 1. Introduction

Septic arthritis is an important disease in horses with potentially grave consequences. The early detection of joint infection, aggressive treatment, and reliable identification of the eradication of infection are important for successful outcomes. Common laboratory methods for the diagnosis and monitoring of the progress of septic arthritis include synovial fluid cytology and bacterial culture [1,2,3]. However, these methods may be unreliable for detecting the eradication of joint infection in horses. This often leads to unnecessarily prolonged antimicrobial therapy, which, in turn, may result in deleterious side effects and the development of antimicrobial resistance [4]. For this reason, there has been a shift toward research aiming to discover better diagnostic methods such as the identification and characterization of biomarkers. Therefore, the overall purpose of our research is to identify potential biomarkers which can be used to detect the point of the eradication of infection in equine septic arthritis.

Biomarkers may be defined as measurable indicators used for disease diagnosis and monitoring treatment response or prognosis [5]. Biomarkers may be any substances measured in biological samples, such as proteins, metabolites, mRNA, etc. There have been a number of studies on the discovery of biomarkers of joint diseases in horses [6]. Most studies have concentrated on protein biomarkers [7], with fewer studies exploring other potential classes of biomarkers such as metabolites [8]. Most of these studies investigated potential biomarkers for the diagnostic detection of equine septic arthritis, but none have investigated biomarkers for the eradication of infection in treated equine cases of septic arthritis, which has been the focus of our research over several years. An experimental model of the eradication of infection in equine septic arthritis was established and it was demonstrated that both the standard cytological diagnostic parameters of synovial fluid (i.e., total nucleated cell count, percentage neutrophils, and total proteins) and serum amyloid A (SAA) are useful diagnostic indicators for the detection of equine septic arthritis, but suboptimal biomarkers for detecting the eradication of infection in equine septic joints subjected to treatment [1,9]. Accordingly, comparative discovery proteomics of synovial fluid from septic arthritis joints with active and eradicated infection was performed and a number of differentially abundant proteins that may represent potential biomarkers for the eradication of infection in septic arthritis cases subjected to treatment were identified [10]. The goal of the current study was to comparatively investigate the transcriptome of synovial fluid from septic arthritis joints with active and eradicated infection and compare it to the previously reported proteome [10]. This initiative was stimulated in part by a recently discovered [11] and validated [12] mRNA biomarker panel, SeptiCyte Lab, used for the clinical differentiation between shock caused by sepsis from other causes of shock in human patients. This work suggested that mRNA can be a potential biomarker for distinguishing infectious versus non-infectious causes of inflammation. As a consequence of this discovery, we decided to investigate if mRNA transcripts in synovial fluid could be used as biomarkers for the eradication of infection in equine septic arthritis cases subjected to therapy. Specifically, the objectives of this study were to (1) perform a transcriptomics analysis of synovial fluid from horses with experimental septic arthritis before and after the eradication of infection and from horses with experimentally induced non-septic synovitis; (2) identify differentially expressed mRNA from synovial fluid in horses with septic arthritis compared to synovial fluid after the eradication of infection and synovial fluid from horses with experimentally induced non-septic synovitis; (3) compare the differentially expressed mRNA to the previously reported proteomics analysis; and (4) compare the differentially expressed mRNA to the mRNA panel SeptiCyte LAB used to distinguish septic shock from non-septic shock in human medicine.

## 2. Materials and Methods

This study was conducted in compliance with the guidelines of the Canadian Council on Animal Care after an appropriate review and approval by the University of Saskatchewan Animal Care and Use Committee and Animal Research Ethics Board (Animal Use Protocol # 20180048). The sample size was determined based on calculations provided by Hart et al. [13]. A sample size of 5 animals per group was determined to be adequate to identify the effect size of 3-fold changes, with a coefficient of variation of 60%, average count of 1000, power of 80%, and *p* < 0.05, which was considered as acceptable. Ten adult American Quarter horses (six mares and four geldings) with a mean ± SD age of 10 ± 3 years (range 7–18 years) and mean ± SD body weight of 528 ± 29 kg (range of 483–580 kg) were used in this experimental study. The horses were considered to be healthy and free of musculoskeletal disease based on a thorough physical examination, complete lameness examination, and complete blood work (complete blood count [CBC], biochemistry profile including measurement of systemic blood SAA). Lameness was subjectively evaluated while the horses were walking in a straight line and a circle and was graded as sound, lame at the walk, or non-weight-bearing lame.

For this experimental study, horses were randomly assigned into septic arthritis (*n* = 5) and experimentally induced non-septic synovitis (*n* = 5) groups. Septic arthritis and non-septic synovitis were, respectively, induced as previously described [1]. Briefly, septic arthritis was induced by the injection of 10^8^ colony forming units of *Escherichia coli* isolated previously from a clinical isolate of equine septic arthritis into a middle carpal joint. Non-septic synovitis was induced by the injection of 5 ng of lipopolysaccharide diluted in in-house sterile phosphate-buffered saline (pH 7.4) into a middle carpal joint. Twenty-four hours post induction (post induction day (PID) 1), the injected joints in both groups were treated by arthroscopic lavage and postoperatively with antimicrobial regional limb perfusion, and intraarticular and systemic antibiotic treatment as previously described [1]. Briefly, arthroscopic lavage was performed using 20 L of saline, with the intra-articular administration of 500 mg of gentamicin before the end of the procedure. Regional limb perfusion was performed with 1 g of gentamicin PID 1, 2, and 3. Systemic therapy included sodium penicillin 22,000 IU/kg, q6h, IV, gentamicin sulfate 6.6 mg/kg, q24h, IV, and phenylbutazone 2.2 mg/kg, q12h, IV for 6 days, followed by trimethoprim-sulfamethoxazole 24 mg/kg, q12h orally for additional 5 days.

The horses in the experimentally induced non-septic synovitis group developed varying degrees of transient lameness, which peaked at 4 h after induction: sound (*n* = 2) and lame at a walk (*n* = 3), All horses were sound at a walk by 18 h after the induction of non-septic synovitis and lameness was not observed in any horse thereafter. All horses’ vital parameters were within normal limits throughout the study period. For the septic arthritis group, all horses developed marked effusion of the middle carpal joint, and presented non-weight-bearing lameness, peaking at 12 h after the inoculation of bacteria. After arthroscopic lavage was performed at 24 h, all horses became sound at a walk and their vital parameters were all within normal limits for the remainder of the study. All horses survived to the end of the study and no lameness was present when they walked in a straight line and in a circle.

Samples of blood and synovial fluid from the left mid-carpal joint (injected) and right mid-carpal joint (non-injected) were collected before the induction of an experimental model (PID 0) and on PID 1, 2, 3, 4, 7, and 10. Synovial fluid and nucleated cell count were performed at the Prairie Diagnostic Services (Saskatoon, Canada) using Advia 2120i (Siemens Healthcare Diagnostics, Tarrytown, New York, USA). Bacterial culture of the synovial fluid was performed as previously described [1]. Briefly, synovial fluid was inoculated into 100 mL of brain–heart infusion (BHI) broth (BD Bacto Brain Heart Infusion; Becton Dickinson, Franklin Lakes, NJ, USA) and incubated aerobically at 36 °C for 3 days. In addition, synovial fluid was also inoculated onto Columbia blood agar and incubated aerobically at 36 °C. If bacterial growth on BHI was detected, subculturing on CBA was performed. Bacterial isolates were identified using MALDI-TOF (Bruker, Billerica, MA, USA) at Prairie Diagnostic Services, according to the manufacturer’s protocol and as described in a previous study [14].

### 2.1. RNA Extraction and Library Preparation

Synovial samples from the septic arthritis group at PID 1 and PID 4 and the experimentally induced non-septic synovitis PID 1 were used for mRNA sequencing. The total RNA was extracted from the horse synovial fluid using Trizol (Thermo Fisher Scientific, Waltham, MA, USA) and then further purified with PureLink RNA Mini Kit (Invitrogen, Thermo Fisher Scientific, Waltham, MA, USA), according to the manufacturer’s instructions. DNA from the samples was removed from the RNA following DNase I treatment (New England Biolabs, Ipswich, MA, USA, Cat: M0303S), as described by the manufacturer, and cleaned using the Monarch RNA Cleanup kit (New England Biolabs, Ipswich, MA, USA). The RNA quality was assessed with the Qubit RNA HS Assay (Thermo Fisher Scientific, Waltham, MA, USA) and RNA Screentape (Agilent Technologies, Santa Clara, CA, USA). Sequencing libraries were generated from 50–100 ng of total RNA using the NEBNext rRNA Depletion Kit v2 (Human/Mouse/Rat; New England Biolabs, Ipswich, MA, USA) and NEBNext Ultra II Directional RNA Library Prep Kit (New England Biolabs, Ipswich, MA, USA) following the manufacturer’s instructions.

### 2.2. Sequencing

Sequencing libraries were evaluated using the Qubit dsDNA HS Assay (Thermo Fisher Scientific, Waltham, MA, USA) and a D1000 Screentape (Agilent Technologies, Santa Clara, CA, USA). The barcoded libraries were pooled equimolar and 75 bp paired-end reads were generated on a NextSeq 550 instrument (Illumina, San Diego, CA, USA).

### 2.3. Data Processing

Sequencing reads were extracted and demultiplexed using bcl2fastq on the BaseSpace version 7.2.0 software service (Illumina, San Diego, CA, USA) with default settings. Sequencing adapters and low-quality bases were trimmed using fastp v0.20.1 [15]. The reads were aligned using STAR v2.7.9a [16] and the Horse assembly reference (EquCab3.0). Reads had to be present in all samples of at least one group to be selected for further analysis.

### 2.4. Differential Gene Expression Analysis

The counts of the mRNA sequencing reads were analyzed using R Bioconductor package Rsubread v2.105 [17]. The Deseq2 v1.22.2 function from the Bioconductor package was used for differential gene expression analysis [18], and an adjusted false discovery rate of *p* < 0.01 was considered to be statistically significant. The further selection of candidate mRNA biomarkers from the differentially expressed genes was performed in Excel v2404 (Microsoft, Redmond, WA, USA) according to the following criteria: (1) the minimal count value in septic arthritis PID1 horses must be at least 3-fold higher compared to the maximal value in eradicated septic arthritis PID4 horses; (2) the minimal value in septic arthritis PID1 horses must be at least 2-fold higher than the maximal value in experimentally induced non-septic synovitis PID1 horses; and (3) the minimal normalized count in all septic arthritis horses at PID1 is >1000 (Figure 1). Figures were prepared in GraphPad Prism version 8.2.1 for Windows (GraphPad Software, La Jolla, CA, USA).

## 3. Results

### 3.1. Experimental Model

In the septic arthritis group, the synovial fluid culture was positive for *E. coli* in all horses at PID 1, and in two out of the five horses at PID2. All samples of synovial fluid from the injected joints at PID3 and after were negative. In the experimentally induced non-septic synovitis group, the culture of the injected joint was negative in all horses. The contamination rate of the bacterial culture was approximately 7% (10/140 of cultures).

In the septic arthritis group at PID 1, the nucleated cell count (NCC) increased to the median of 214.8 × 10^9^/L (range 178.2–254.6 × 10^9^/L) and decreased steadily until PID10 (Figure 2).

On PID4 in the septic arthritis group, the median nucleated cell count was 3.6 × 10^9^/L (range 0.2–11.4 × 10^9^/L), and was lower than the septic arthritis threshold of 30 × 10^9^/L [19] in all horses, supporting the eradication of infection by PID4 in the septic arthritis group. In the experimentally induced non-septic synovitis, the presence of inflammation was also confirmed by the nucleated cell count in the synovial fluid, which reached a median of 133.4 × 10^9^/L (range 55.1–133.4 × 10^9^/L). The non-injected contralateral control joints in both septic arthritis and experimentally induced non-septic synovitis did not yield a positive culture result. One horse in the septic arthritis group had a moderate NCC increase at PID1 (19 × 10^9^/L) and one horse in the experimentally induced non-septic synovitis group had a marked NCC at PID2 (83 × 10^9^/L), which decreased to 23.4 × 10^9^/L at PID3 and 3.9 at PID4. In the absence of a positive bacterial culture, these NCC increases were attributed to transient non-septic reactions to manipulations and sample collection [20].

Based on the bacterial culture and nucleated cell count, the eradication of infection was established by PID4 in all experimental horses (hereafter, eradicated septic arthritis at PID4). Synovial fluid samples from the septic arthritis group at PID4 were used for the transcriptomics analysis of samples post eradication of infection.

### 3.2. Transcriptomics Analysis

The transcriptomics analysis of the synovial fluid in all samples revealed 30,371 total genes. Principal component analysis showed clustering within treatment groups (Figure 3).

To identify potential mRNA biomarkers for the eradication of infection, the following criteria were employed (Figure 1). Genes which were not present in all samples of at least one group were excluded from further analysis, leaving 12,071 genes. Then, 6247 differentially expressed genes between the septic arthritis at PID1 and eradicated septic arthritis at PID4 groups and 2216 genes between the septic arthritis at PID1 and experimentally induced non-septic synovitis groups (FDR adjusted *p* < 0.01) were identified.

The next step in identifying putative mRNA biomarkers was to select those differentially expressed mRNA transcripts in which the minimal count value of samples with septic arthritis was 3-fold higher than the maximal count value of samples post-eradication of infection. Twenty-five mRNA transcripts satisfied this criterion. Out of these 25 genes, 9 genes were selected which were also differentially expressed between the septic arthritis group at PID1 and experimentally induced non-septic synovitis group at PID1, and the minimal count value in the septic arthritis group at PID1 was 2-fold higher than the maximal value in the experimentally induced non-septic synovitis group at PID1. Finally, differentially expressed mRNAs in the septic arthritis group with a minimal normalized count > 1000 were selected, which yielded eight upregulated genes satisfying all criteria (Figure 4; Table 1). Similar criteria were used to identify differentially expressed genes downregulated in the septic arthritis group at PID1 compared to the eradicated septic arthritis group at PID4 and experimentally induced non-septic synovitis. No genes meeting these criteria were found.

Out of the eight upregulated genes meeting our criteria, two protein products were identified in our previously published discovery proteomics experiment [10], namely interleukin-1 receptor accessory protein and glycogenin-1 (Table 2). However, these proteins did not exhibit a statistically significant difference in abundance in the horses with septic arthritis compared to the horses post eradication of joint infection and the horses with experimentally induced non-septic synovitis in the previous proteomic study.

All 26 differentially abundant synovial fluid proteins which were previously identified in our proteomics study [10] had corresponding mRNA transcripts identified in this experiment. Seven of the twenty-six mRNAs were upregulated, two were downregulated (*p* < 0.01), and seventeen were not significantly different when comparing the synovial fluid from horses with septic arthritis at PID1 and eradicated septic arthritis at PID4 (before and after the eradication of infection). When comparing the septic arthritis group at PID1 and experimentally induced non-septic synovitis group at PID1, seven mRNAs were upregulated, one was downregulated, and eighteen were not significantly different. Neutrophil collagenase, BTB/POZ domain-containing protein KCTD12, ubiquitin-like modifier-activating enzyme 1, and guanine nucleotide-binding protein subunit beta-2 were upregulated when comparing septic arthritis at PID1 to both eradicated septic arthritis at PID4 and experimentally induced non-septic synovitis at PID1 (Table 3). Although none of the mRNA transcripts corresponding to the previously reported differentially abundant proteins satisfied the rigorous criteria for the differentially expressed mRNA used in this study, mRNA for neutrophil collagenase (MMP1/8) and interleukin-1 receptor antagonist protein (IL1RN) stand out as being consistently upregulated in the synovial fluid from joints with septic arthritis.

The transcriptomics results were also compared with the mRNA transcripts that constitute SeptiCyte Lab, namely PLAC8, PLA2G7, LAMP1, and CEACAM4 [11]. SeptiCyte Lab test produced a score which was calculated based on the threshold cycle numbers for the four mRNA transcripts mentioned above. PLAC8 and LAMP1 gave a positive contribution to the score (upregulated), while PLA2G7 and CEACAM4 gave a negative contribution (downregulated) [11]. In our data, PLAC8 (two subunits, PLAC8A and PLAC8B), LAMP1, and PLA2G7 were identified, while CEACAM4 was not identified (Figure 5).

These mRNA transcripts were not differentially expressed between the groups (*p* > 0.01). However, LAMP1, PLAC8A, and PLAC8B tended to be higher in the septic arthritis group at PID1 compared to experimentally induced the non-septic synovitis group, while PLA2G7 tended to be lower in septic arthritis compared to non-septic synovitis (Figure 5), which is consistent with the pattern of the mRNA expression measured by the SeptiCyte Lab.

## 4. Discussion

In this study, eight putative mRNA biomarkers for the eradication of infection in an equine experimental septic arthritis model were identified. All eight mRNA transcripts were differentially expressed and upregulated in the synovial fluid of horses with septic arthritis at PID1 compared to the same horses post eradication of infection at PID4 and horses with experimentally induced non-septic synovitis at PID1. Some of these mRNA transcripts have been shown to play a role in inflammation and/or infection and sepsis. Thus, glutamine synthetase (GLUL) was found to be important in reducing endotoxin-induced sepsis [21]. GLUL was increased in macrophages in vitro in response to lipopolysaccharide treatment [22]. Interleukin 1 receptor type 2 (IL1R2) is a decoy receptor; an increased expression of IL1R2 leads to reduced interleukin 1 signaling [23]. Serum IL1R2 was found to be significantly elevated in mice injected with inactive *E. coli* and *Staphylococcus aureus*, as well as in septic human patients [24]. IL1R2 was also upregulated in human neonates with bacterial infection [25] and infants with late-onset sepsis [26]. In our previous discovery proteomics study [10], interleukin 1 receptor antagonist protein (IL1RN) was identified as being upregulated in the synovial fluid of horses with septic arthritis compared to horses post eradication of joint infection and horses with experimentally induced non-septic synovitis. The mRNA transcript for this protein was also found in this experiment, and although the normalized counts were slightly higher in horses with septic arthritis, they did not reach our threshold of adjusted *p* value for the differentially expressed proteins. Still, it is interesting to note that both IL1R2 and IL1RN share similar biological function of inhibiting IL-1 signaling, suggesting that this pathway may be valuable biomarker for distinguishing septic from non-septic inflammation. Interleukin 1 receptor accessory protein (IL1RAP) is required for the transmission of interleukin 1 signaling [27]. Glycogenin 1 (GYG1) mRNA was significantly upregulated in the peripheral blood of children with pneumococcal meningitis [28]. The other four transcripts were either non-specific (PTK2B) or did not correspond to known proteins, with protein existence inferred from homology. The potential biological significance of these mRNA transcripts requires further investigation. Nevertheless, the 3- to 11-fold higher count of the eight mRNA transcripts in the synovial fluid of the septic arthritis horses at PID1 compared to the same horses with eradicated septic arthritis at PID4 warrants further investigation of these mRNA transcripts as putative biomarkers for the eradication of infection in clinical cases of equine septic arthritis. This is supported even more by the 2- to 7-fold higher counts of these mRNA transcripts in the synovial fluid from horses with septic arthritis at PID1 compared to the horses with experimentally induced non-septic synovitis at PID1.

Comparing the transcriptomics data obtained in this experiment to our previously published discovery proteomics data [10], all mRNA transcripts corresponding to the 26 differentially abundant proteins identified in the discovery proteomics study were found [10]. Four mRNA transcripts were upregulated in the synovial fluid with septic arthritis. Of these, neutrophil collagenase (MMP8) was most consistent with a fold change of 22.9 when comparing septic arthritis before (PID1) and post eradication of joint infection (PID4), and a 4.2 fold change when comparing septic arthritis and experimentally induced non-septic synovitis PID1 (Table 3). This follows the pattern of the nucleated cell count. Interestingly, interleukin-1 receptor antagonist protein (IL1RN) was upregulated in the septic arthritis joints compared to the experimentally induced non-septic synovitis (fold change 9.3, *p* < 0.01), and was upregulated in the synovial fluid with septic arthritis before the eradication of infection (PID1) compared to post eradication of infection (PID4), with the adjusted *p*-value being close to the significance cut-off used in this study (fold change 4.2; *p* = 0.014). This result provides support for the interleukin 1 signaling pathway perhaps providing useful biomarkers for distinguishing septic arthritis from experimentally induced non-septic synovitis, as well as detecting the eradication of infection. However, neither MMP8 nor IL1RN mRNA transcripts met our criteria of the >3-fold difference between the minimal count value in the horses with septic arthritis at PID1 and horses post eradication of septic arthritis at PID4, and the >2-fold difference between the minimal count value in the horses with septic arthritis at PID1 and horses with experimentally induced non-septic synovitis at PID1, and are, therefore, not included in the list of putative mRNA biomarkers. The lack of greater correspondence between the transcriptomics and proteomics results may be attributed to a number of factors, such as differences in the samples (centrifuged synovial fluid for proteomics versus whole synovial fluid for transcriptomics), the post-transcriptional regulation of mRNA, posttranslational modification of proteins, different sources of proteins within the synovial fluid (for example, proteins in synovial fluid may be produced in liver or other remote organs and subsequently leaked into joint as was demonstrated previously) [9], and individual biologic variation in our experiments.

SeptiCyte Lab is a validated assay based on a panel of four mRNA transcripts, which is used to distinguish between septic shock and non-septic shock [12]. Since our objective was similar despite differences in the species, disease, and substrate tested, we wanted to compare our results with this panel. Three of the four transcripts used in SeptiCyte Lab were found in our data. In our data, these transcripts were not significantly differentially expressed in the synovial fluid with septic arthritis compared to post eradication of infection. Interestingly, there was significant upregulation of PLAC8 in septic arthritis compared to experimentally induced non-septic synovitis, which is consistent with the behavior of this mRNA transcript within the SeptiCyte Lab panel. The lack of correspondence between SeptiCyte Lab and our data may have been due to differences in the inflammatory response in target species, as well as differences in the mRNA profile between the whole blood used for SeptiCyte Lab and the equine synovial fluid used in this study.

## 5. Conclusions

In this study, a transcriptomics analysis of equine synovial fluid in horses with septic arthritis, post eradication of joint infection, and in horses with experimentally induced non-septic synovitis was performed. Eight most promising mRNA targets as potential biomarkers of the eradication of joint infection in horses were identified. However, in interpreting these data, it is important to remember the limitations of our study. The number of animals per group was relatively low, contributing to variability in the data. The experimental animals were horses, which were not controlled for biological variation. Experimental models of septic arthritis and experimentally induced non-septic synovitis were used, which are not representative of clinical cases of equine septic arthritis. The effects of the treatment itself could alter the mRNA expression in synovial fluid, which was not evaluated in this study. Nevertheless, our study provides a novel transcriptomics foundation for the discovery of biomarkers of the eradication of joint infection in horses, and demonstrates that mRNA may be a viable target for such biomarkers.

## Figures and Tables

**Figure 1 vetsci-11-00299-f001:**
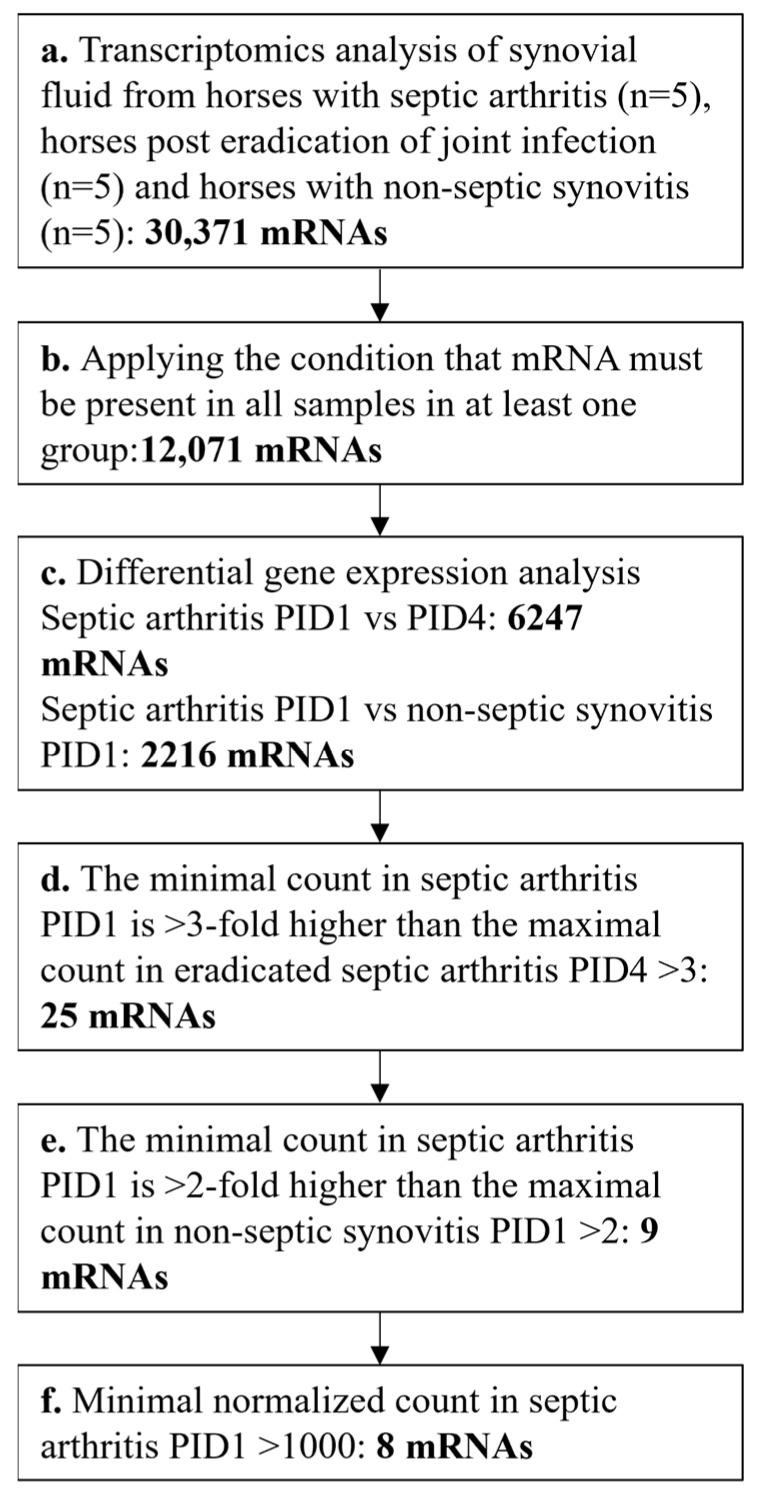
Flowchart demonstrating selection criteria for the differentially expressed mRNA transcripts. (**a**) Total number of mRNA gene transcripts identified. (**b**) mRNA transcripts which were not present in all samples of at least one group were rejected. (**c**) Differential gene expression was performed using Deseq2. (**d**) mRNA transcripts were selected if the minimal count in septic arthritis PID1 was at least 3-fold higher compared to the maximal count in eradicated septic arthritis PID4. (**e**) mRNA transcripts were selected if the minimal count in septic arthritis PID1 was at least 2-fold higher compared to the maximal count in experimentally induced non-septic synovitis PID1. (**f**) For the upregulated mRNA transcripts in septic arthritis PID1 minimal normalized count >1000 was selected.

**Figure 2 vetsci-11-00299-f002:**
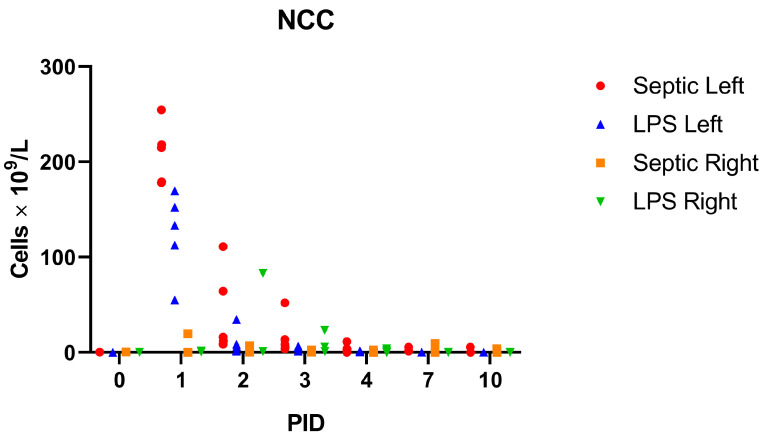
Nucleated cell count (NCC) in synovial fluid from the left (injected with *E. coli*) and right (non-injected contralateral control) joints of horses with septic arthritis (Septic Left ● and Septic Right ■) and horses with LPS-induced experimentally induced non-septic synovitis (LPS Left ▲ injected with LPS and LPS Right ▼ non-injected contralateral control).

**Figure 3 vetsci-11-00299-f003:**
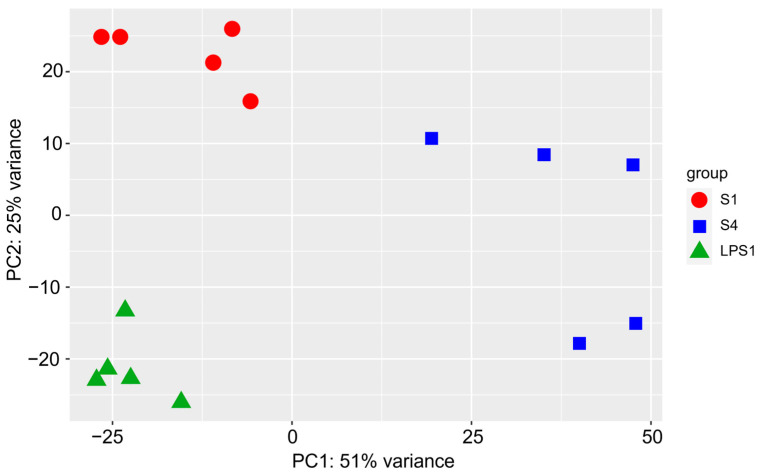
Principal component analysis showing clustering within groups, based on total normalized gene count. S1 ●: septic arthritis at PID 1; S4 ■: eradicated septic arthritis at PID 4; LPS1 ▲: experimentally induced non-septic synovitis at PID 1. PID: post induction day.

**Figure 4 vetsci-11-00299-f004:**
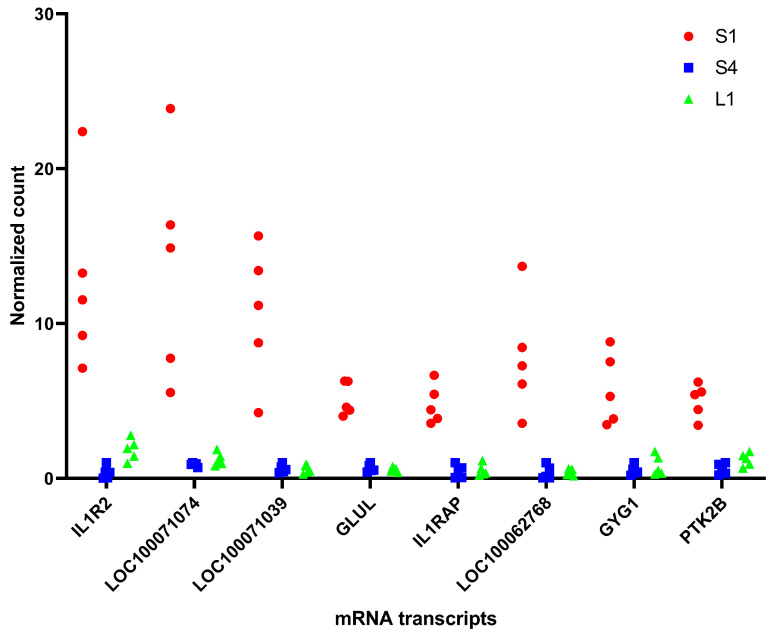
Putative mRNA biomarkers for eradication of infection in equine septic arthritis model. Differentially expressed mRNA transcripts (listed in Table 1), that met selection criteria used in this study. The counts were normalized to the maximal value in horses post eradication of infection (eradicated septic arthritis at PID 4). S1 ●: septic arthritis at PID 1; S4 ■: eradicated septic arthritis at PID 4; L1 ▲: experimentally induced non-septic synovitis at PID 1. PID: post induction day.

**Figure 5 vetsci-11-00299-f005:**
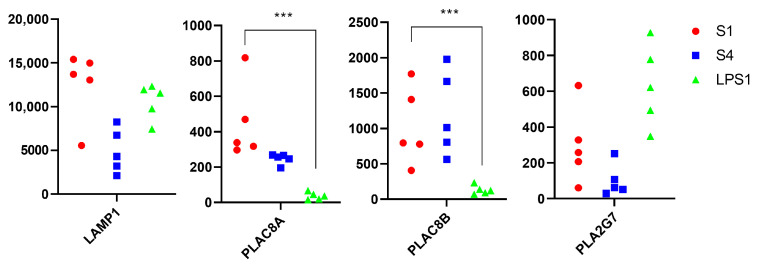
mRNA transcripts which are used in the SeptiCyte Lab biomarker panel, as detected in our study. Raw counts are graphed on *y*-axis. PLAC8 was identified by subunits A and B. There is significant upregulation of PLAC8A and PLAC8B in septic arthritis compared to experimentally induced non-septic synovitis. S1●: septic arthritis at PID1; S4■: eradicated septic arthritis at PID4; LPS1▲: experimentally induced non-septic synovitis at PID1. *** indicates *p* < 0.001 for differential gene expression results performed using Deseq2.

**Table 1 vetsci-11-00299-t001:** Putative mRNA biomarkers for eradication of infection in equine septic arthritis model (see also Figure 4). Differentially expressed mRNA transcripts that met selection criteria in synovial fluid of horses with septic arthritis at PID1 vs. eradicated septic arthritis at PID4 (post eradication of infection) and experimentally induced non-septic synovitis PID1.

Gene Name	Gene ID	UniProt Accession Number	Protein Name
IL1R2	ENSECAG00000000288	F7DK71	Interleukin-1 receptor type 2
LOC100071074	ENSECAG00000039739	F6QB61	Peptidoglycan-recognition protein
LOC100071039	ENSECAG00000032321	F7DQG1	Peptidoglycan-recognition protein
GLUL	ENSECAG00000015865	F6TAZ0	Glutamine synthetase
IL1RAP	ENSECAG00000005083	F7AWQ9	Interleukin 1 receptor accessory protein
LOC100062768	ENSECAG00000006255	F7ADT4	G-protein coupled receptors family 1 profile domain-containing protein
GYG1	ENSECAG00000023192	F6U6H7	Glycogenin 1
PTK2B	ENSECAG00000018924	F7CGD4	Non-specific protein-tyrosine kinase

**Table 2 vetsci-11-00299-t002:** Proteins corresponding to the putative mRNA biomarkers, identified in the discovery proteomics study [10]. Mean and standard deviation of mass spectra intensities for proteins is reported.

Gene Name		Septic Arthritis PID1	Eradicated Septic Arthritis PID4	Non-Septic Synovitis PID1
IL1RAP	Mean	614,082	623,448	1,358,139
	SD	408,114	241,337	1,041,416
GYG1	Mean	0	0	30,143
	SD	0	0	45,269

**Table 3 vetsci-11-00299-t003:** Transcriptomics results compared to the previously published discovery proteomics results [10]. mRNA transcripts identified in this study corresponding to the previously reported 26 differentially abundant proteins, with statistical significance (*p* value) and fold change indicated as calculated during differential expression analysis using Deseq2.

Protein Name	Gene ID	Gene Name	*p*-Value Septic Arthritis PID1 vs. Eradicated Septic Arthritis PID4	Fold Change	*p*-Value Septic Arthritis PID1 vs. Non-Septic Synovitis PID1	Fold Change
E3 ubiquitin-protein ligase TRIM9	ENSECAG00000003278	TRIM9	0.082	−4.1	0.859	1.3
DCC-interacting protein 13-alpha	ENSECAG00000017473	APPL1	0.130	−1.3	0.031	−1.5
Ubiquitin-like-conjugating enzyme ATG3	ENSECAG00000011954	ATG3	0.586	−1.1	0.849	1.1
Aldehyde dehydrogenase family 16 member A1	ENSECAG00000023437	ALDH16A1	0.145	1.3	0.638	−1.1
Serine/threonine-protein kinase 24	ENSECAG00000007682	STK24	0.007	1.5	0.604	1.1
Protein ABHD14B	ENSECAG00000022421	ABHD14B	0.700	−1.1	0.359	−1.2
Transforming protein RhoA	ENSECAG00000033791	RHOA	0.009	1.6	0.032	1.5
Guanine nucleotide-binding protein subunit beta−2	ENSECAG00000017221	GNB2	0.000	2.4	0.000	1.9
Ubiquitin-conjugating enzyme E2 L3	ENSECAG00000032131	UBE2L6	0.170	1.6	0.000	7.1
*N*-acetyltransferase ESCO1	ENSECAG00000016120	ESCO1	0.000	2.8	0.077	1.6
Twinfilin-2	ENSECAG00000014047	TWF2	0.068	1.3	0.003	1.5
Nuclear transport factor 2	ENSECAG00000021530	NUTF2	0.000	−2.7	0.048	−1.8
Elongation factor 1-gamma	ENSECAG00000014334	EEF1G	0.000	−2.3	0.841	1.1
Septin-7 (Fragment)	ENSECAG00000022693	SEPTIN7	0.216	−1.2	0.473	1.2
GTP-binding nuclear protein Ran	ENSECAG00000020532	RAN	0.071	−1.5	0.283	1.3
Synaptic vesicle membrane protein VAT-1 homolog	ENSECAG00000016700	VAT1	0.790	1.1	0.001	−2.4
Alpha-1-acid glycoprotein 2	ENSECAG00000036760	NA	0.656	−1.3	0.126	−2.2
BTB/POZ domain-containing protein KCTD12	ENSECAG00000002083	KCTD12	0.000	3.6	0.002	2.8
Interleukin-1 receptor antagonist protein	ENSECAG00000027864	IL1RN	0.014	4.2	0.000	9.3
Ubiquitin-like modifier-activating enzyme 1	ENSECAG00000014002	UBA1	0.000	2.6	0.000	2.8
Heterogeneous nuclear ribonucleoprotein D0	ENSECAG00000022692	HNRNPDL	0.051	−1.3	0.678	1.1
Neutrophil collagenase	ENSECAG00000023733	MMP1/8	0.000	22.9	0.000	4.2
Fermitin family homolog 3	ENSECAG00000008107	FERMT3	0.815	1.1	0.332	1.3
Coronin-7	ENSECAG00000024788	CORO1A	0.769	1.1	0.026	1.8
Poly(rC)-binding protein 1	ENSECAG00000024218	PCBP4	0.020	−7.0	0.547	−2.0
Myotrophin	ENSECAG00000024284	MTPN	0.927	−1.0	0.229	1.3

## Data Availability

The datasets analyzed during the current study are available from the corresponding author on reasonable request.

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
