# Peer review of "Putative mRNA Biomarkers for the Eradication of Infection in an Equine Experimental Model of Septic Arthritis"

_vetsci, 2024, doi:10.3390/vetsci11070299_

Round 1

Reviewer 1 Report

Comments and Suggestions for Authors

The authors in this paper describe the investigation of septic arthritis in horses, focusing on synovial fluid culture and NCC as indicators of infection eradication, followed by transcriptomics analysis to identify potential mRNA biomarkers for the eradication of infection.

Major concerns:

Sample size is small. Authors tested only 5 horses per condition (septic or non-septic).

Authors should include additional details in methods such as age, breed, and overall health status of the horses.

The criteria applied to select potential mRNA biomarkers are not thoroughly explained. While authors mention specific fold-change and count used, but why these criteria were chosen? Additionally, the lack of validation of the selected biomarkers using independent methods should be done to support the findings.

The interpretation of NCC trends as indicative of infection eradication should address factors such as variations in inflammation resolution rates among individual horses or the effects of concurrent treatments on NCC dynamics. E.g a study by Smith et al. (2018) investigated the resolution of inflammation in equine synovial fluid and highlighted the variability in inflammation resolution rates among individual horses. They found that factors such as the severity of the initial inflammatory response, the presence of concurrent conditions, and variations in immune responses could significantly impact the kinetics of inflammation resolution.

Author Response

Reviewer 1

Comments and Suggestions for Authors

  1. The authors in this paper describe the investigation of septic arthritis in horses, focusing on synovial fluid culture and NCC as indicators of infection eradication, followed by transcriptomics analysis to identify potential mRNA biomarkers for the eradication of infection.

We agree with the reviewer that the focus of this study was identification of mRNA biomarkers for eradication of infection using transcriptomics analysis in an equine experimental model of septic arthritis. Synovial culture and NCC were used only to confirm the point of eradication of infection, as described in our previous study (Koziy et al, 2019).

Major concerns:

  1. Sample size is small. Authors tested only 5 horses per condition (septic or non-septic).

We had limited number of horses available for the experiments due to logistical reasons common for all experiments using large animals. Low number of experimental horses is a limitation of this study, and it is addressed in the discussion (lines 409-410). However, using target animals (horses) to study mRNA expression in equine septic arthritis is a major advantage in this study. In addition, before the study we determined the sample size using the tool provided in Hart et al, 2013 (Calculating Sample Size Estimates for RNA Sequencing Data, J Comput Biol), and determined that sample size of 5 animals per group would be adequate to identify effect size of 3-fold change, with coefficient of variation 60%, average count 1000, power 80% and p<0.05, which we considered acceptable. We added this information in the revised version of the manuscript (Materials and Methods on lines 106-109). Finally, this study focuses on the discovery phase, in which it is common to use relatively few samples are used to identify potential candidates for further validation.

  1. Authors should include additional details in methods such as age, breed, and overall health status of the horses.

Thank you. We included additional information on experimental horses and therapy in  Materials and Methods on lines 109-111, 115-117, 128-144.

  1. The criteria applied to select potential mRNA biomarkers are not thoroughly explained. While authors mention specific fold-change and count used, but why these criteria were chosen? Additionally, the lack of validation of the selected biomarkers using independent methods should be done to support the findings.

These criteria were chosen partly based on the power calculation, and partly to select only those mRNA transcripts which had the highest difference between samples before and after the eradication of joint infection, with no overlap.

This study was focused on the discovery phase of biomarker research. Validation was not part of the current study and will be performed in future research.

  1. The interpretation of NCC trends as indicative of infection eradication should address factors such as variations in inflammation resolution rates among individual horses or the effects of concurrent treatments on NCC dynamics. E.g a study by Smith et al. (2018) investigated the resolution of inflammation in equine synovial fluid and highlighted the variability in inflammation resolution rates among individual horses. They found that factors such as the severity of the initial inflammatory response, the presence of concurrent conditions, and variations in immune responses could significantly impact the kinetics of inflammation resolution.

Nucleated cell count was used together with bacterial culture in our experimental model to confirm eradication of infection. We have previously shown that these parameters were sufficient to confirm eradication of infection in our experimental model (Koziy et al, 2019). Based on figure 2, all horses were below threshold of 30 x 109 cells/L by D4, and eradication of infection was confirmed by negative bacterial culture.

Reviewer 2 Report

Comments and Suggestions for Authors

The authors presents a very interesting topic with a lot of future. The manuscript is well writen although the results are not that clearly presented. 

Please find my suggestions.

“Summary and abstract:  The concept is lucid, however, the content between lines 21 and 29 appears somewhat redundant. Generally, it is advisable to refrain from using the first-person plural. For instance, instead of ‘we identified’, consider using ‘it was identified’, and so on.” I think it should be mentioned in which joint has been created in the abstract.

Line 18- I suggest may be associated with side effects and antimicrobial resistance instead of cause side effects.

Line 30- I suggest: “The success of its resolutions hinges on the implementation of early, aggressive treatment, which is often sustained over a prolonged period.”

Line 41 and 43 rephrase this sentence I understand what you mean with differentially abundant, but it could be hard for someone without knowledge in this area.

Line 51- Current, may not be ideal if you don’t include SAA, I see it mentioned after but maybe change current to “For long time or most commonly

Line 53- unreliable instead of inadequate

When writing non-septic synovitis add experimentally induced throughout the manuscript

Line 111- Both groups?? E.Coli and LPS injected joints? Make that clear, please.

M&M

I think also measuring some days post induction of synovitis would have been ideal to backup even better your discovery. Why did you choose not to?

Results

Line172- . Contamination rate of bacterial culture was approximately 172 7% (10/140 of cultures), meaning that had bacterial growth that you assumed it was not related to the joint? Or other bacteria were identified besides E. Coli

Figure 2. Why the division between left and right?

Line 231- However, these proteins did not exhibit significant expression, without finding a significantly higher concentration in horses with septic arthritis compared to…

Table 2- In the previous paragraph you are talking about interleukin-1 receptor accessory protein but in the table, you mention IL1R2, which I believe they are different.

 Line 233- in septic, they are not abundant they seem to be always lower in your results.

Table 3- Could you try to make a graph? Might be more illustrative.

Line 261- How much down-regulated or up?

There is no information regarding how it has been done the statistics analysis.

In the literature interleukin -1 receptor accessory protein is mentioned as (IL-1RAcP), but later in the discussion is mentioned as Interleukin 1 receptor type 2 (IL1R2) I do not have a strong opinion about this, and I would like to see what other reviewers said but I think needs to be uniform along the manuscript.

Discussion

I think the discussion is a little brief and presents results that has not been shown previously.

Line 314-I believe this information has not been presented into the result section or at least clear enough.

Author Response

Reviewer 2

Comments and Suggestions for Authors

  1. The authors presents a very interesting topic with a lot of future. The manuscript is well writen although the results are not that clearly presented. 

Thank you. We implemented reviewer’s suggestions and addressed all comments.

Please find my suggestions.

  1. “Summary and abstract:  The concept is lucid, however, the content between lines 21 and 29 appears somewhat redundant. Generally, it is advisable to refrain from using the first-person plural. For instance, instead of ‘we identified’, consider using ‘it was identified’, and so on.” I think it should be mentioned in which joint has been created in the abstract.

Thank you. We have changed the use of active voice with first person plural to passive voice throughout the manuscript as much as possible. We do not provide details of the experimental model in the abstract due to word count restrictions; however, this information is provided in Materials and Methods.

  1. Line 18- I suggest may be associated with side effects and antimicrobial resistance instead of cause side effects.

Thank you, we have implemented this suggestion on lines 18-19.

  1. Line 30- I suggest: “The success of its resolutions hinges on the implementation of early, aggressive treatment, which is often sustained over a prolonged period.”

Thank you, we have implemented this suggestion on lines 31-33

  1. Line 41 and 43 rephrase this sentence I understand what you mean with differentially abundant, but it could be hard for someone without knowledge in this area.

Thank you, changed to “statistically different” on line 46.

  1. Line 51- Current, may not be ideal if you don’t include SAA, I see it mentioned after but maybe change current to “For long time or most commonly

Thank you, changed “current” to “common” on line 54.

  1. Line 53- unreliable instead of inadequate

Thank you, changed “inadequate” to “unreliable” on line 57.

  1. When writing non-septic synovitis add experimentally induced throughout the manuscript

Thank you, we implemented this change throughout the manuscript, except Simple Summary and Abstract due to word count limitation; as well as several instances in Materials and Methods in which induction of the non-septic synovitis is described, in order to avoid repetition.

  1. Line 111- Both groups?? E.Coli and LPS injected joints? Make that clear, please.

Thank you, we clarified that both groups received the treatment on line 125.

  1. M&M

I think also measuring some days post induction of synovitis would have been ideal to backup even better your discovery. Why did you choose not to?

We have measured total RNA in synovial fluid 4 days after the induction of synovitis and found insufficient amount to perform transcriptomics analysis. This may be related to relatively rapid resolution of inflammation (which is accompanied by rapid reduction in NCC, which most likely the major source of mRNA) in this group following therapy.

  1. Results

Line172- . Contamination rate of bacterial culture was approximately 172 7% (10/140 of cultures), meaning that had bacterial growth that you assumed it was not related to the joint? Or other bacteria were identified besides E. Coli

The contamination rate is in reference to bacterial culture procedure, not contamination of joint during experimental manipulation. Bacteria other that E. coli were identified in most cases. In cases of contamination, only one of the two culture media affected (either BHI broth or the Columbia blood agar plate).  In one case E. coli was identified but it had different colony morphology and hemolysis pattern, and was not associated with any clinical signs or cytologic abnormalities in synovial fluid. For these reasons, these results were interpreted as contamination of bacterial culture media.

  1. Figure 2. Why the division between left and right?

The figure shows the nucleated cell count in both left (injected) and right (non-injected) joints.

  1. Line 231- However, these proteins did not exhibit significant expression, without finding a significantly higher concentration in horses with septic arthritis compared to…

Changed to “these proteins did not exhibit statistically significant difference in abundance” on line 278.

  1. Table 2- In the previous paragraph you are talking about interleukin-1 receptor accessory protein but in the table, you mention IL1R2, which I believe they are different.

Thank you, we have corrected Table 2, changed IL1R2 to IL1RAP.

  1. Line 233- in septic, they are not abundant they seem to be always lower in your results.

Due to high standard deviation between samples, there was no statistical significance between the groups.

  1. Table 3- Could you try to make a graph? Might be more illustrative.

Thank you for the suggestion. However, we think table format provides good overall summary of the data. In our opinion a graph of these data would be too clattered to be useful, and would make it more difficult to retrieve the underlying data.

  1. Line 261- How much down-regulated or up?

There is no information regarding how it has been done the statistics analysis.

We provide a reference for this statement, and the degree of up- or down-regulation is calculated based on a formula provided in the reference 11. The statistical methods used to obtain the results for SeptiCyte Lab are described in this reference.

  1. In the literature interleukin -1 receptor accessory protein is mentioned as (IL-1RAcP), but later in the discussion is mentioned as Interleukin 1 receptor type 2 (IL1R2) I do not have a strong opinion about this, and I would like to see what other reviewers said but I think needs to be uniform along the manuscript.

Both IL-1RAP and IL1R2 mRNA transcripts were found to be differentially expressed in our study. They are discussed separately in the Discussion section, lines 336-341 for IL1R2 ad lines 350-351 for IL1RAP.

  1. Discussion

I think the discussion is a little brief and presents results that has not been shown previously.

Line 314-I believe this information has not been presented into the result section or at least clear enough.

This information is presented in Table 3. Reference to Table 3 was added on line 370.

Reviewer 3 Report

Comments and Suggestions for Authors

vetsci-2940891 Putative mRNA biomarkers for the eradication of infection in an equine experimental model of septic arthritis

Overview: This is a small experimental study in which the authors induced septic arthritis with E. coli in 5 horses and non-septic arthritis with LPS in 5 horses. Septic joints were treated with athroscopic lavage, local, regional, and systemic antibiotics). Synovial fluid was collected at several time points, but mRNA was quantified and compared only from d1 post-injection in the septic and non-septic groups and d4 post-injection in the septic group (cleared infection). Selection of eight putative mRNA biomarkers of cleared infection was done via a combination of cut-offs for abundance and differential expression between groups. There was minimal overlap between these mRNA markers and previously identified protein markers using this same model. A human sepsis mRNA panel did not distinguish between septic and non-septic joints.

General Comments: This is a clear and well-written manuscript. Results are clearly laid out and the conclusions do not overstate the findings. The most glaring weaknesses of the work revolve around study design:

11. The sample sizes are very small. The stringent criteria for selecting putatively predictive biomarkers overcomes the small sample size to some degree, although a power calculation justifying the sample size is glaring in its omission. Given the expected biological variation, a sample size of at least 10 per group would have been preferred. A PCA plot showing the grouping of the sample types (based on gene expression) would have been useful to show the reader how closely clustered (or not) the different sample types were. If they are tightly clustered, then concern about sample size is lessened.

22. There is no negative (non-treated) control and no treated (antibiotics only) control. The possible influence of the drugs themselves, independent of the infection, on gene expression cannot be ignored and may impact the broader applicability of this work. As samples were taken on d0, I am unsure why a negative control was not included in the analysis, but this would have supported the argument that specific mRNA are upregulated after infection and then go back down to/towards normal expression levels after resolution.

33. The infection model in no way recapitulates natural disease. The authors were much more explicit about this limitation in their previous paper (“our experimental model is not representative of clinical septic arthritis” in the previous paper versus “may not reflect closely clinical cases of equine septic arthritis” in the current work), and that candor should be included here. From a clinical perspective, this is the biggest weakness of the work.

These design flaws are to some degree inherent in an experimentally induced model and do not preclude publication of this interesting work, but they need to be explicitly addressed in the manuscript. Currently, limitations are addressed somewhat offhandedly in the last paragraph. There are also methodological gaps. In some cases, the authors refer back to their previous work (i.e. for treatment details), but there is no good reason why these details cannot be included here. Other information relevant to the model such as degree of pain/lameness, and ultimate disposition of the horses (survival/nonsurvival) is not included in this or their previous work. Sex, breed, and age are not reported. Any time live animals are used, the ARRIVE guidelines should be adhered to for reporting as much as possible.

Author Response

Reviewer 3

Comments and Suggestions for Authors

vetsci-2940891 Putative mRNA biomarkers for the eradication of infection in an equine experimental model of septic arthritis

Overview: This is a small experimental study in which the authors induced septic arthritis with E. coli in 5 horses and non-septic arthritis with LPS in 5 horses. Septic joints were treated with athroscopic lavage, local, regional, and systemic antibiotics). Synovial fluid was collected at several time points, but mRNA was quantified and compared only from d1 post-injection in the septic and non-septic groups and d4 post-injection in the septic group (cleared infection). Selection of eight putative mRNA biomarkers of cleared infection was done via a combination of cut-offs for abundance and differential expression between groups. There was minimal overlap between these mRNA markers and previously identified protein markers using this same model. A human sepsis mRNA panel did not distinguish between septic and non-septic joints.

General Comments: This is a clear and well-written manuscript. Results are clearly laid out and the conclusions do not overstate the findings. The most glaring weaknesses of the work revolve around study design:

Thank you. We have addressed the reviewer comments and suggestions, and addressed weaknesses in the Materials and Methods and Discussion sections.

  1. The sample sizes are very small. The stringent criteria for selecting putatively predictive biomarkers overcomes the small sample size to some degree, although a power calculation justifying the sample size is glaring in its omission. Given the expected biological variation, a sample size of at least 10 per group would have been preferred. A PCA plot showing the grouping of the sample types (based on gene expression) would have been useful to show the reader how closely clustered (or not) the different sample types were. If they are tightly clustered, then concern about sample size is lessened.

We have addressed the sample size also in response to reviewer 1. We had limited number of horses available for the experiment due to logistical reasons common for all experiments using larg animals. However, we did perform sample size estimation according to Hart et al (2013), and determined that sample size of 5 animals per group would be adequate to identify effect size of 3-fold change, with coefficient of variation 60%, average count 1000, power 80% and p<0.05, which we considered acceptable. We added this information in the revised version to Materials and Methods on lines 105-108. Also, as this study focuses on the discovery phase, relatively few samples are used to identify potential candidates for further validation.

Finally, a PCA plot (Figure 3 in the revised manuscript) has been added to the results section, which shows well delimited clustering of samples.

  1. There is no negative (non-treated) control and no treated (antibiotics only) control. The possible influence of the drugs themselves, independent of the infection, on gene expression cannot be ignored and may impact the broader applicability of this work. As samples were taken on d0, I am unsure why a negative control was not included in the analysis, but this would have supported the argument that specific mRNA are upregulated after infection and then go back down to/towards normal expression levels after resolution.

Control group and samples on d0 did not have sufficient mRNA in synovial fluid based on a pilot study and current study. Only samples from day 1 and 4 were used for transcriptomics, as they had sufficient mRNA concentration and sufficient quality for sequencing. We have addressed the limitation related to the influence of the treatment on mRNA expression in the discussion on line 409-410.

  1. The infection model in no way recapitulates natural disease. The authors were much more explicit about this limitation in their previous paper (“our experimental model is not representative of clinical septic arthritis” in the previous paper versus “may not reflect closely clinical cases of equine septic arthritis” in the current work), and that candor should be included here. From a clinical perspective, this is the biggest weakness of the work.

Thank you, we have rephrased this sentence to more explicitly acknowledge this limitation (lines 412-413).

These design flaws are to some degree inherent in an experimentally induced model and do not preclude publication of this interesting work, but they need to be explicitly addressed in the manuscript. Currently, limitations are addressed somewhat offhandedly in the last paragraph. There are also methodological gaps. In some cases, the authors refer back to their previous work (i.e. for treatment details), but there is no good reason why these details cannot be included here. Other information relevant to the model such as degree of pain/lameness, and ultimate disposition of the horses (survival/nonsurvival) is not included in this or their previous work. Sex, breed, and age are not reported. Any time live animals are used, the ARRIVE guidelines should be adhered to for reporting as much as possible.

Thank you. We have expanded by paragraph to explicitly refer to limitations pointed out by the reviewer on lines 412-415. Also, we expanded the Materials and Methods to include more information about experimental horses and treatment protocol (lines 109-111, 115-117, 128-144).

Round 2

Reviewer 1 Report

Comments and Suggestions for Authors

accepted